# Landslide Dynamic Susceptibility Mapping Base on Machine Learning and the PS-InSAR Coupling Model

**Fasheng Miao** [1], **Qiuyu Ruan** [1], **Yiping Wu** [1,*], **Zhao Qian** [1], **Zimo Kong** [2] and **Zhangkui Qin** [1]

[1]   Faculty of Engineering, China University of Geosciences, Wuhan 430074, China; fsmiao@cug.edu.cn (F.M.);
     qiuyuruan@cug.edu.cn (Q.R.)
[2]   College of Artificial Intelligence, Southwest University, Chongqing 400715, China
*   Correspondence: ypwu@cug.edu.cn; Tel.: +86-13971100757

**Abstract:** Complex and fragile geological conditions combined with periodic fluctuations in reservoir water levels have led to frequent landslide disasters in the Three Gorges Reservoir area. With the development of remote sensing technology, many scholars have applied it to landslide susceptibility assessment to improve model accuracy; however, how to couple these two to obtain the optimal susceptibility assessment model remains to be studied. Based on Sentinel-1 data, relevant data, and existing research results, the information value method (IV), random forest (RF), support vector machine (SVM), and convolutional neural network (CNN) models were selected to analyze landslide susceptibility in the urban area of Wanzhou. Models with superior performance will be coupled with PS-InSAR deformation data using two methods: joint training and weighted overlay. The accuracy of different models was assessed and compared with the aim of determining the optimal coupling model and the role of InSAR in the model. The results indicate that the accuracy of different landslide susceptibility prediction models is ranked as RF > SVM > CNN > IV. Among the coupled dynamic models, the performance ranking was as follows: InSAR jointly trained RF (IJRF) > InSAR weighted overlay RF (IWRF) > InSAR jointly trained SVM (IJSVM) > InSAR weighted overlay SVM (IWSVM). Notably, the IJRF model, which combines InSAR deformation data through joint training, exhibited the highest accuracy, with an AUC value of 0.995. In the factor importance analysis within the IJRF model, InSAR deformation data ranked third after hydrological distance (0.210) and elevation (0.163), with a value of 0.154. A comparison between landslide dynamic susceptibility mapping (LDSM) and landslide susceptibility mapping (LSM) revealed that the inclusion of InSAR deformation data effectively reduced false positives around the landslide areas. The results suggest that joint training is the most suitable coupling method, allowing for the optimal expression of InSAR deformation data and enhancing the predictive accuracy of the model. This study serves as a reference for future research and provides a foundation for landslide risk management.

**Keywords:** landslide; dynamic susceptibility mapping; PS-InSAR; coupling model



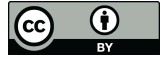

## 1. Introduction

Landslides are a prevalent geologic disaster worldwide [1,2]. Reservoir water storage can significantly change the hydrogeological conditions of bank slopes, leading to landslides and bank slope deformation [3–5]. As one of the largest reservoirs, numerous landslides have developed in the Three Gorges Reservoir area [6], posing a serious threat to the life safety of local residents [7–9]. As an important means for landslide risk control, landslide susceptibility assessment analyzes the nonlinear relationship between landslides and related environmental factors [10], providing an accurate spatial probability distribution of landslide susceptible areas [11,12]. This assessment holds great significance for landslide disaster prevention and mitigation efforts.

The key to landslide susceptibility assessment lies in the construction of a susceptibility assessment model, the calculation of the spatial probability distribution of the

landslide susceptible area [10], and the conduction of landslide susceptibility mapping (LSM). Three types of landslide susceptibility assessment models are commonly used in existing studies, including heuristic models, mathematical statistical models, and machine learning (ML) models. Heuristic models such as the analytic hierarchy process (AHP) [13] require researchers to judge the weights and thresholds of the factors based on their expertise [14]. However, this subjective experience leads to the uncertainty of the model [15]. Mathematical statistical models [16] including the information value method (IV) [17] and deterministic coefficient method, among others, rely on the engineering analogy method and superimpose factors in different ways to express the nonlinear relationship between factors and landslides. Machine learning models such as logistic regression [18], SVM [19], and RF [16] can efficiently capture the relationship between factors and landslides, which are widely used in landslide susceptibility assessment on account of their excellent performance and efficient modeling process [20], although partial models mat be challenging to interpret due to the black-box analysis process. Deep learning models, represented by CNN, being developed and derived from machine learning models [14], are able to mine deep features and relationships in data through their neural network structure, enabling them to address complex problems [21,22]. Although different models exhibit advantages in different aspects, there is lack of consensus on the optimal model for predicting landslide susceptibility [23]. To verify the applicability and performance of different models, four models were employed for landslide susceptibility prediction research in this study: IV from the mathematical statistical models, SVM and RF from the machine learning models, and CNN from the deep learning models.

The evolution of a landslide often initiates with the creep stage [24]; monitoring ground deformation can be used to determine the extent and stability of landslides. [3,25,26]. However, the current landslide susceptibility assessment typically constructs models through static factors without considering dynamic features such as ground deformation, which leads models to misjudge certain landslide areas [23]. With the development of remote sensing technology, the accurate and long-time surface deformation monitoring data from InSAR (interferometric synthetic aperture radar) [27–29] enables the construction of a landslide dynamic susceptibility model [30,31]. Utilizing the persistent scatterer interferometry (PSI) technique, Mishra and Jain [32] retrieved the displacements of the Baglihar Dam Reservoir slope. Hussain et al. [33] updated the landslide inventory for susceptibility mapping. Zhu et al. [34] constructed the landslide dynamic susceptibility assessment model by means of an empirical matrix. Liu et al. [35] and Cao et al. [36] used the weighted overlay method to construct a landslide dynamic susceptibility assessment model which achieved LDSM (landslide dynamic susceptibility mapping) by weighting the InSAR deformation data and LSM. Various landslide susceptibility assessment models and different InSAR coupling methods were used in these studies to prove the significance of InSAR deformation data in landslide susceptibility assessment. However, it remains uncertain which coupling method can be used to obtain the best landslide dynamic susceptibility assessment model.

In summary, this study aimed to compare the performance of different coupled models, find the optimal method for updating the LSM with InSAR deformation data, and obtain the best landslide dynamic susceptibility assessment model. The study in this paper was conducted in three stages: (1) Based on the landslide-related factor data in the study area, we constructed IV, SVM, RF, and CNN models to obtain the LSM; validation assessment and performance comparison of the selected models indicated that RF and SVM exhibit the best performance. (2) Using weighted overlay and joint training methods, the SVM and RF models are coupled with the InSAR deformation data to construct dynamic susceptibility models and generate the LDSM. (3) We compared the accuracy of the applied dynamic susceptibility models based on various statistical indicators, appraised the applicable coupling method, and analyzed the role of the InSAR deformation data in model through LDSM.

## 2. Study Area and Dataset

### 2.1. Study Area

The study area is in the urban area of Wanzhou District, within the Three Gorges Reservoir area. Its geographic coordinates range from 108°19′50″E to 108°31′10″E and from 30°43′20″N to 30°51′30″N (Figure 1), covering an approximate area of 289 km². Wanzhou is situated in the Yangtze River valley of the eastern Sichuan Basin, characterized by a subtropical monsoon climate with features of a warm and humid climate, abundant rainfall, four distinct seasons, and notable vertical stratification [37]. It experiences an average annual temperature of 18.2 °C and an annual precipitation of 1155.8 mm. The region belongs to the Yangtze River system and features an intricate network of rivers with significant elevation differences and a dendritic distribution.

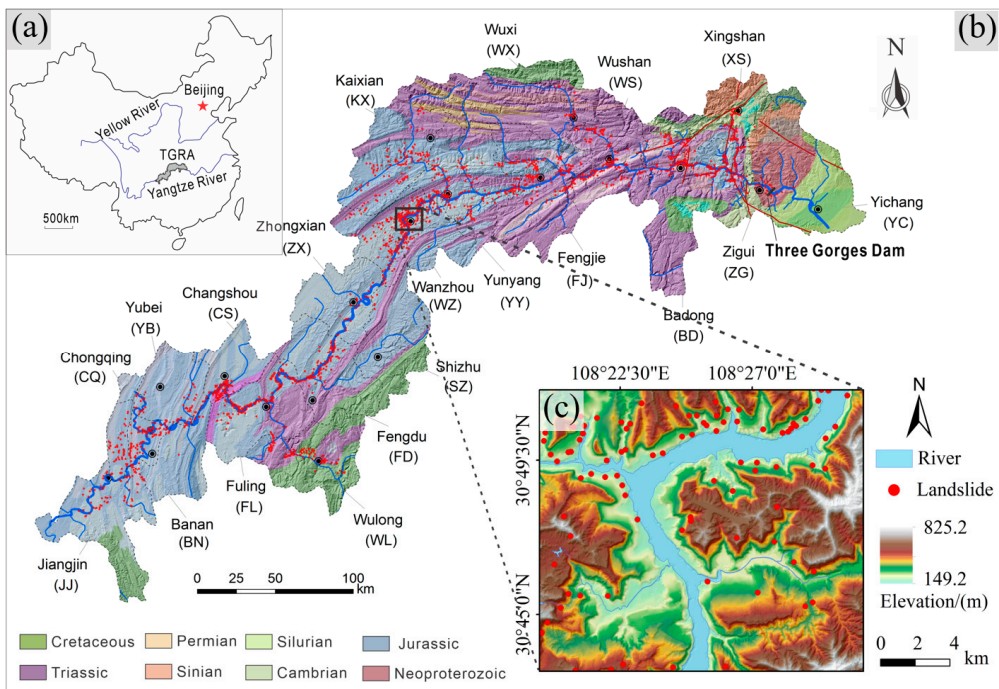

**Figure 1.** (**a**) Geographical location of the Three Gorges Reservoir area; (**b**) Geological map and geological hazards (red dots) in the Three Gorges Reservoir area; (**c**) DEM of study area.

Within the study area, there is a distribution of erosional and depositional landforms along the Yangtze River and its tributaries, characterized by floodplains and terraces [38], in which the overall elevation is relatively low, with the lowest altitude at 149.2 m above sea level. The surrounding area consists of low mountains and hills, with elevations mostly in the range of 440~620 m and reaching a maximum of 825.2 m.

The outcropping strata in the study area belong to the monoclinic Middle Jurassic period and Quaternary [39]. The $J_{2s}$ lithology comprises sandstone, siltstone, silty claystone, and mudstone, accounting for approximately 73% of the total; the $J_{3s}$ lithology consists of purple–red mudstone, muddy siltstone, and brownish-red siltstone with varying thicknesses, interbedded with purple–gray fine-grained feldspathic sandstone, making up around 23% of the total area. Additionally, small quantities of $J_{3p}$, $J_{2xs}$, $J_{2x}$ are distributed in the eastern and southeastern parts of the study area, as shown in Figure 2f.

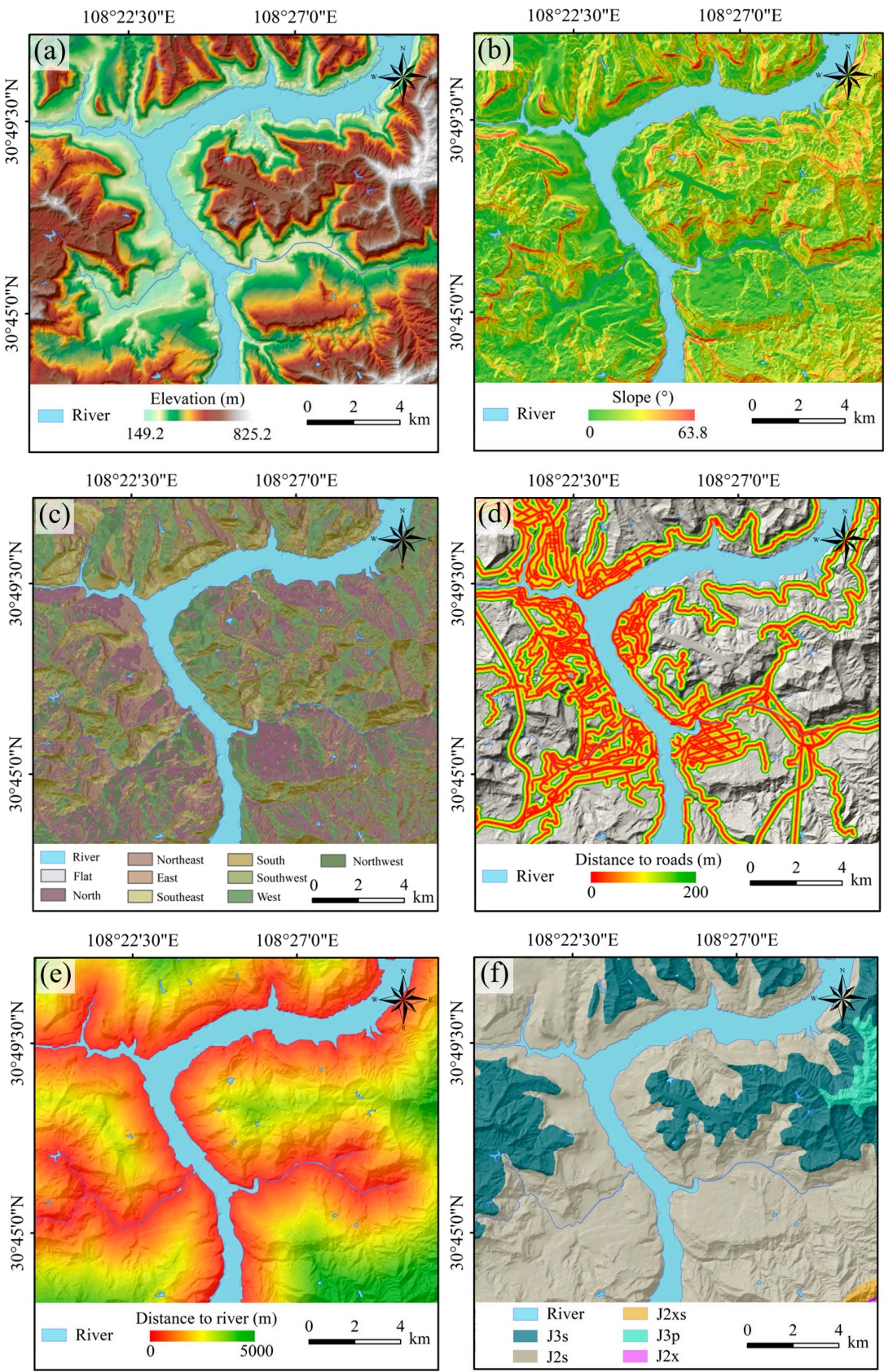

**Figure 2.** *Cont.*

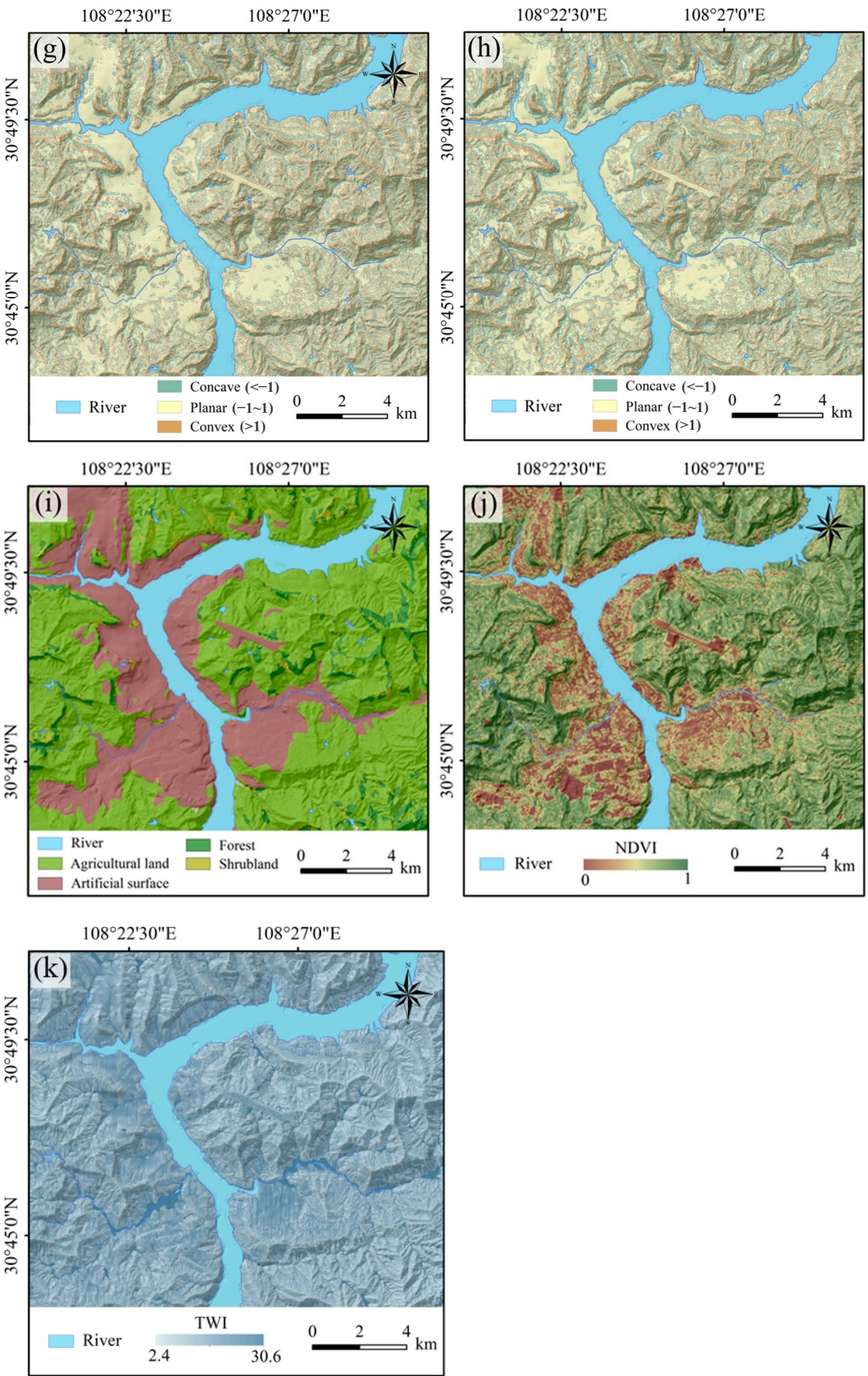

**Figure 2.** Landslide-related factors in the study area: (**a**) elevation, (**b**) slope, (**c**) aspect, (**d**) distance to roads, (**e**) distance to waters, (**f**) lithology, (**g**) plan curvature, (**h**) profile curvature, (**i**) land use type, (**j**) NDVI, and (**k**) TWI.

## 2.2. SAR Dataset

The Sentinel-1A SAR C-band data were obtained from the European Space Agency (ESA). The SAR dataset comprises 24 ascending scenes and each image covers an area of approximately 40 km × 85 km along the Wanzhou–Yunyang section of the Yangtze River (as shown in Figure 3), with a total coverage area of 3380.8 km². The image acquisition time spans from 17 July 2019 to 12 June 2021. Detailed data acquisition dates are included in Table 1.

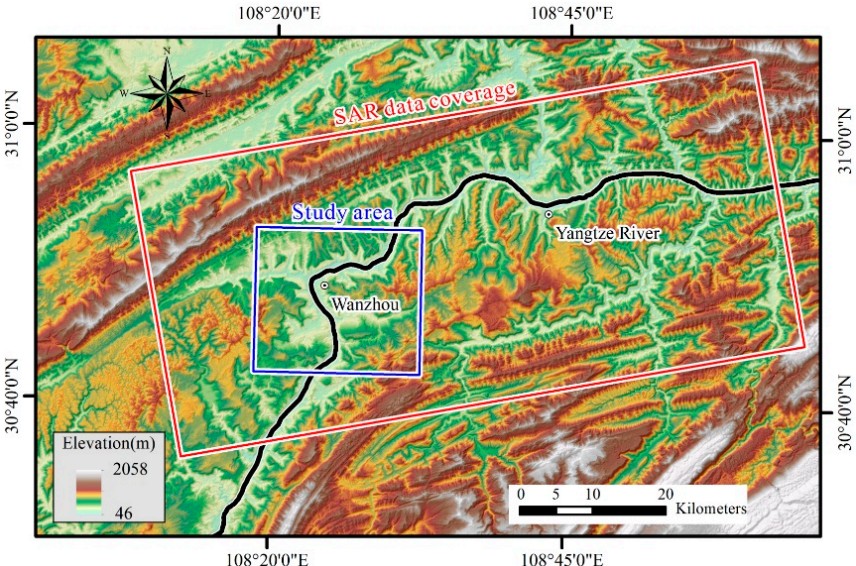

**Figure 3.** Sentinel-1A SAR data coverage and study area wrapped on the 12.5 m DEM used for the PS processing.

**Table 1.** Detailed data acquisition dates of Sentinel-1A.

| No. | Date (yyyy-mm-dd) | No. | Date (yyyy-mm-dd) |
|---|---|---|---|
| 1 | 2019-07-17 | 13 | 2020-07-11 |
| 2 | 2019-08-10 | 14 | 2020-08-16 |
| 3 | 2019-09-15 | 15 | 2020-09-21 |
| 4 | 2019-10-09 | 16 | 2020-10-15 |
| 5 | 2019-11-14 | 17 | 2020-11-08 |
| 6 | 2019-12-08 | 18 | 2020-12-02 |
| 7 | 2020-01-13 | 19 | 2021-01-07 |
| 8 | 2020-02-18 | 20 | 2021-02-12 |
| 9 | 2020-03-13 | 21 | 2021-03-08 |
| 10 | 2020-04-18 | 22 | 2021-04-13 |
| 11 | 2020-05-12 | 23 | 2021-05-07 |
| 12 | 2020-06-17 | 24 | 2021-06-12 |

The SAR images are single-look complex data generated using dual HH+HV polarization and the interferometric wide (IW) swath mode, with an incidence angle of 34° and a spatial resolution of 5 × 20 m. Additionally, the 12.5 m DEM covering the study area was prepared for the PS processing.

## 2.3. Landslide-Related Factors and Dataset

This paper considers various conditional factors related to landslide occurrence combined with existing studies on landslide susceptibility and available data in the study area [6,38,39]. A basic dataset for landslide susceptibility assessment in the study area was constructed from aspects of stratum lithology, topography, and geological structure [40].

This dataset comprises eleven factors: elevation, slope, aspect, distance to roads, distance to river, lithology, plan curvature, profile curvature, NDVI, TWI, and land use type.

The digital elevation model (DEM) serves as the primary tool for representing topographic features. Topographic and geomorphic factors including elevation, slope, aspect, plan curvature, profile curvature, and TWI are derived from the DEM data. These factors provide valuable insights into the topographic relief, convexity, and other characteristics which are significant causative factors for landslide occurrence [14].

Water flow can accelerate soil erosion and weaken soil strength [41], while vehicular traffic on roads and surrounding structures can exert additional loads [42]. Collectively, these factors contribute to the risk of slope instability. Therefore, the distance to the river and the distance to roads were calculated by Euclidean distance tools in ArcGIS based on vector data of rivers and roads in the study area and subsequently incorporated into the dataset.

Geological lithology factors are extracted from geological maps of the study area. Using Landsat 5/8 remote sensing imagery from the Google Earth Engine (GEE) platform, spatial data with a resolution of 30 m for NDVI and land use types can be obtained. Vegetation loss due to human activities is one of the major factors causing shallow landslides [14]. NDVI and land use types provide insights into vegetation density and human activity in the study area [43].

Standardizing all factors to a consistent spatial scale, cropping them to the study area's extent, and dividing them into grid of 30 × 30 m with a total of 434,005 uniform raster cells, we obtained a raster dataset for the study area which can be used for further landslide susceptibility analysis. During the training process for machine learning and deep learning, the dataset was divided: 70% was used for the training set and 30% was used for the validation set. A total of 10,000 positive and 20,000 negative samples were randomly selected from the training set and used for training landslide susceptibility assessment models so that the susceptibility prediction could be conducted for the whole study area.

## 3. Methodology

### 3.1. Multicollinearity Analysis

In landslide susceptibility assessment, multicollinearity of influencing factors will lead to redundancy and the mutual interference of inputs, which has a negative influence on the predictive ability of the model [36]. To assess the independence of factors, researchers often employ methods such as principal component analysis, the variance inflation factor (VIF), regression analysis, and the Pearson correlation coefficient [14,44–46]. In this study, VIF and TOL (tolerance) are used to measure the linear correlation of factors and can be calculated using Equation (1).

$$VIF = \frac{1}{1-R^2} = \frac{1}{TOL} \tag{1}$$

Here, $R^2$ represents the variance between influencing factors. When VIF > 5 or TOL < 0.2, the multicollinearity in the dataset will be considered.

### 3.2. Landslide Susceptibility Assessment Models

#### 3.2.1. Information Value (IV) Model

The information value (IV) method is a mathematical statistical analysis method based on information theory and the engineering analogy method [16]. It was proposed by Claude Shannon in the 1940s during his study of information theory and has since been widely applied in the assessment of geohazard susceptibility. The occurrence of geohazards is influenced a multitude of factors, with each factor contributing to varying degrees [47]. The IV model constructs the landslide susceptibility assessment model by analyzing the distribution relationship between the known hazard points and factors, quantitatively

calculating the weight of the factor on the occurrence of the hazard, and standardizes the scales of each factor [13,48–50]. The formula is as follows:

$$I = \sum_{i=1}^{n} I_i = \sum_{i=1}^{n} \ln \frac{N_i/N}{S_i/S} \tag{2}$$

where: $I$ represents the information value of the assessment raster cells; $I_i$ is the information value provided by factor $i$ for geohazards; $N_i$ is the number of raster cells in which geohazards have occurred for influence factor $i$; $N$ is the total number of geohazard raster cells in study area; $S_i$ is the total number of raster cells for influence factor $i$; and $S$ is the total number of raster cells in the study area.

### 3.2.2. Random Forest (RF)

The random forest (RF) model is an integrated learning model based on decision trees [50]. It constructs multiple decision trees to make individual predictions and combines the results by a majority vote to arrive at the final prediction [49,51]. Leveraging bootstrap aggregating and the random subspace method, the RF method randomly samples training data for each decision tree and selects a subset of features for node splitting [14]. This random selection process can control both the bias and variance, reduce inter-feature correlations, and enhance the model's robustness against outliers and noise [52]. Additionally, RF offers advantages such as not requiring dimensionality reduction for high-dimensional problems and the ability to evaluate the importance of each feature in classification tasks [53].

### 3.2.3. Support Vector Machine (SVM)

The support vector machine (SVM) model, introduced by Vapnik et al. [54], is a machine learning model that maps data into high-dimensional feature space using kernel functions and identifies the optimal hyperplane for data classification with the aim of maximizing the distance between data samples [14,19,24]. Based on support vector theory and structural risk minimization, it can be applied to address issues such as classification, regression, and pattern recognition [55]. The most generalized kernel function is the Gaussian kernel (also known as the radial basis function, RBF) [51]; based on the RBF, the final decision function can be composed as follows:

$$g(x) = sign\left(\sum_{i=1}^{n} \gamma_i \alpha_i K(x_i, x_j) + b\right) \tag{3}$$

where $K(x_i, x_j)$ is the RBF kernel function. Since its inception, SVM has rapidly emerged as an indispensable tool in the field of machine learning; its exceptional classification capabilities, adaptability, and interpretability have led to its widespread application across various domains.

### 3.2.4. Convolutional Neural Network (CNN)

The convolutional neural network (CNN) model is a deep learning model initially proposed by LeCun for addressing image recognition tasks [56]. It is a NN (neural network) architecture model consisting of convolution layers, pooling layers, and fully connected layers [57]. By performing convolution operations with appropriately shaped kernels, transmitting and mapping data through the neurons, CNN can extract features from data progressively. The convolution algorithm is as follows:

$$C_j = \sum_{i=0}^{N} (w_j x_b + b_j) \quad (j = 1, 2, \cdots, k) \tag{4}$$

Here, $k$ represents the number of convolution kernels; $C_j$ represents the output of the $j$-th convolution kernel; $i$ signifies the spatial position; $x_i$ signifies the input within the convolution window; $w_j$ refers the weights; $b_j$ refers the bias. CNNs gradually im-

prove their model performance to fit training data and accomplish classification tasks through multiple iterations by calculating the loss function, backpropagating gradients, and updating weights.

In this study, a CNN model was constructed for landslide susceptibility assessment, comprising six convolution layers, six pooling layers, and two fully connected layers. Max-pooling layers were added to reduce the feature dimensions of the convolution layer outputs, aiding the model in learning different features and deep relationships from the input data [58]. Subsequently, the learned features were combined, transformed, and mapped to various categories through the fully connected layers. Adding the ReLU and tanh functions as activation functions for the convolutional and fully connected layers enables nonlinear transformations of the output results, thereby enhancing the model's ability to express intricate relationships [59]. The ReLU and tanh functions are as follows:

$$f(x) = \begin{cases} x & if \ x > 0 \\ 0 & if \ x \le 0 \end{cases} \quad \max(0, x) \tag{5}$$

$$f(x) = tanh(x) = \frac{e^x - e^{-x}}{e^x + e^{-x}} \tag{6}$$

Finally, the model combines the extracted features and maps them to the (0, 1) interval through the sigmoid activation function, generating probability values as the predictions results for landslide classification.

### 3.3. Model Accuracy Verification Methods

We employed ROC (receiver operating characteristic) curves, a confusion matrix, and the FR (frequency ratio) to quantitatively evaluate the accuracy and performance of models. The ROC curve is one of the most used accuracy evaluation methods in probabilistic models. The curve is constructed using specificity and sensitivity based on the model's prediction results and evaluates the model accuracy through the AUC (area under the ROC curve) [60]. AUC values fall within the interval of [0.5, 1], and a higher AUC value approaching 1 indicates a better model accuracy, while a value near 0.5 suggests the model lacks predictive capability [61].

The confusion matrix categorizes binary classification results into four classes [36]: true positive (TP), true negative (TN), false positive (FP), and false negative (FN). Based on the parameters derived from the confusion matrix, the following four model performance evaluation metrics can be calculated [14]:

Accuracy measures the proportion of correctly classified samples in the total sample, with the formula:

$$Accuracy = \frac{TP + TN}{TP + FP + TN + FN} \tag{7}$$

Recall assesses the model's ability to identify positive samples, represented by the proportion of correctly predicted positive samples out of all true positive samples, and can be calculated using:

$$Recall = \frac{TP}{TP + FN} \tag{8}$$

Precision is the proportion of correctly predicted positive samples out of all predicted positive samples, measuring the accuracy of the model's predictions for positive samples. Its formula is as follows:

$$Precision = \frac{TP}{TP + FP} \tag{9}$$

F1 score is a composite metric that considers both recall and precision, computed as the harmonic mean of both. It is used to comprehensively evaluated the performance of the classification model and its formula is defined as:

$$F1 \ Score = 2 \times \frac{Precision \cdot Recall}{Precision + Recall} \tag{10}$$

The frequency ratio (FR) represents the ratio of the landslide area in a specific attribute interval of a factor to the total area of that attribute interval within the entire study area [10,62]. In different susceptibility partitions generated by various models, calculating their corresponding FR values can indict the partition's ability to express positive and negative landslide samples [13,49]. The formula is as follows:

$$FR = \frac{N_i/N}{S_i/S} \tag{11}$$

FR values greater than 1 indicate that the corresponding attribute interval is more conducive for landslide occurrence and vice versa.

### 3.4. PS-InSAR

PS-InSAR employs interferometry synthetic aperture radar technology in conjunction with persistent scatterers (PS) to monitor subtle surface deformations [63,64]. Based on time-series InSAR data and the single master interferogram method, we conduct operations of registration, clipping, and noise reduction on multiple PS points within the image area of study, resulting in the generation of time-series deformation data for these PS points [65]. Subsequently, an interpolation method was applied to generate a ground deformation map (GDM) by integrating the PS points within the study area. This map provides precise measurements of surface subsidence, uplift, and displacement, thus serving as a valuable basis for the timely identification of potential geological hazards [66].

### 3.5. Dynamic Evaluation of Landslide Susceptibility

The application of InSAR technology in landslide susceptibility assessment enhances the consideration of dynamic factors such as ground uplift and subsidence, enabling the identification of potential landslide areas during the creep phase, leading to more accurate assessment results [67]. Common methods for integrating InSAR deformation data with landslide susceptibility models include joint training [23], weighted overlays [36], and constructing evaluation matrices [35,64]. In this study, we employed the first two coupling methods, selecting the top-performing two models to couple with InSAR deformation data in IV, SVM, RF, and CNN, resulting in dynamic landslide susceptibility assessment models. Subsequently, we calculated the probability distribution of landslide occurrence based on these coupled models, generated the LDSM, and conducted a comparative analysis of model performance.

The joint training method integrates InSAR deformation data as influencing factors into the base dataset. After undergoing the same preprocessing as other factors, InSAR deformation data is jointly trained in the model. This process extracted features from InSAR deformation data by leveraging the nonlinear function of models, revealing its relationship with landslide disasters and constructing a susceptibility assessment model that considers dynamic features.

The weighted overlay method considers both geological conditions and landslide evolution. It assigns weights to the GDM derived from InSAR data and the LSM obtained from landslide susceptibility assessment models after normalizing the absolute values of InSAR deformation data [36]. These two weighted components are overlaid to create a landslide dynamic susceptibility assessment model. The weighted overlay formula is as follows:

$$LDSM = 0.61 * LSM + 0.39 * GDM \tag{12}$$

The flow chart of the research process is shown in Figure 4.

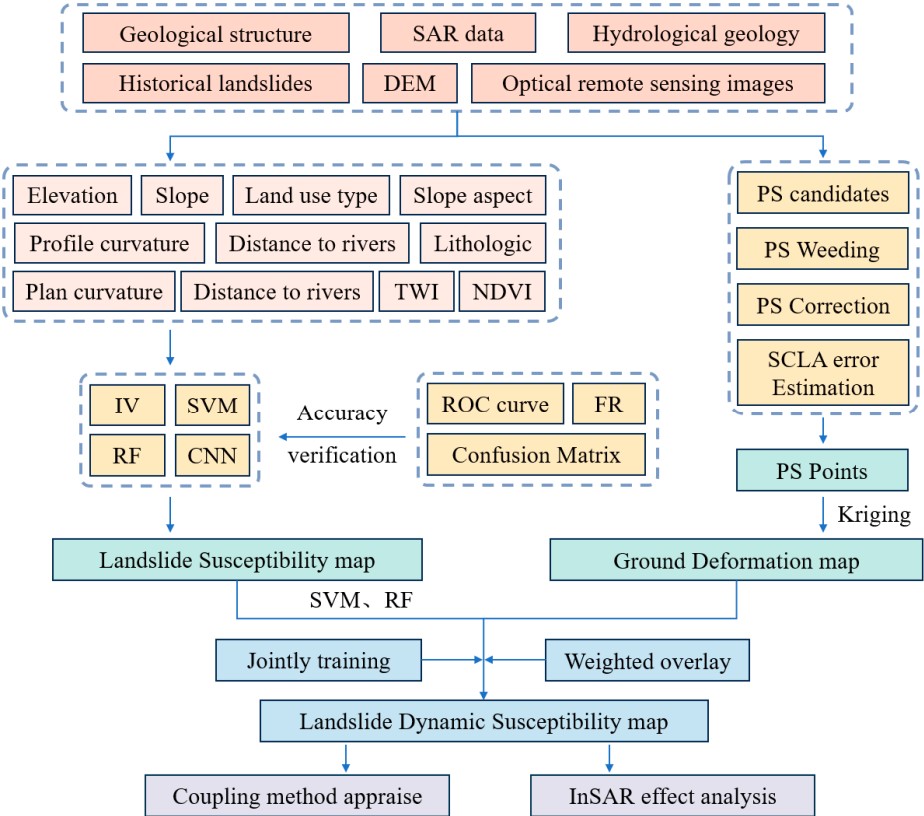

**Figure 4.** Flow chart of the proposed method.

## 4. Results

### 4.1. The Multicollinearity Analysis of Related Factors

Table 2 presents the results of multicollinearity testing among the 11 factors related to landslides. Among all the factors, elevation has the highest VIF value (3.011) and the lowest TOL value (0.332), while the distance to roads displays the lowest VIF value (1.080) and the highest TOL value (0.926). The results of the multicollinearity test show that the VIF and TOL values of selected factors are better than the critical value of insignificant multicollinearity (VIF > 5 and TOL < 0.2), indicating the absence of significant multicollinearity. This suggests that the selected factors have good independence and can be used for training models to achieve highly accurate predictive results.

**Table 2.** Results of multicollinearity assessment (VIF and tolerance value).

| Impact Factors | VIF | TOL |
|---|---|---|
| Elevation | 3.011 | 0.332 |
| Slope | 1.755 | 0.570 |
| Slope aspect | 1.084 | 0.923 |
| Plan curvature | 1.210 | 0.827 |
| Profile curvature | 1.113 | 0.898 |
| Lithologic | 1.186 | 0.843 |
| Distance to rivers | 2.064 | 0.484 |
| Distance to roads | 1.080 | 0.926 |
| TWI | 2.126 | 0.470 |
| NDVI | 2.443 | 0.409 |
| Land use type | 1.829 | 0.547 |

### 4.2. Landslide Susceptibility Mapping

Based on the prepared dataset, the IV, SVM, RF, and CNN methods were applied to assess landslide susceptibility in the study area. Segmented into high (H), moderate (M),

low (L), and very low (VL) landslide susceptibility using the natural breakpoint method, the probability predictions of selected models were visualized as LSM in ArcGIS (Figure 5). As depicted in Figure 5, the H susceptibility areas are primarily concentrated along the Yangtze River and its tributaries, especially in the sloped areas along the banks of the Yangtze River in Tangjiao Village and Tiancheng District, the regions along the bank of the Zhuxi River, and the valley areas at the origins of the Wuqiao River and Longbao River in the southern part of the study area. These areas are distributed along riverbanks and transitional zones between low hills and riverbanks. The erosion caused by fluctuations in reservoir water levels has diminished the stability of slopes in these areas. Furthermore, as these regions are interconnected with densely populated areas, frequent engineering activities and the depletion of vegetation exacerbate the susceptibility to landslides.

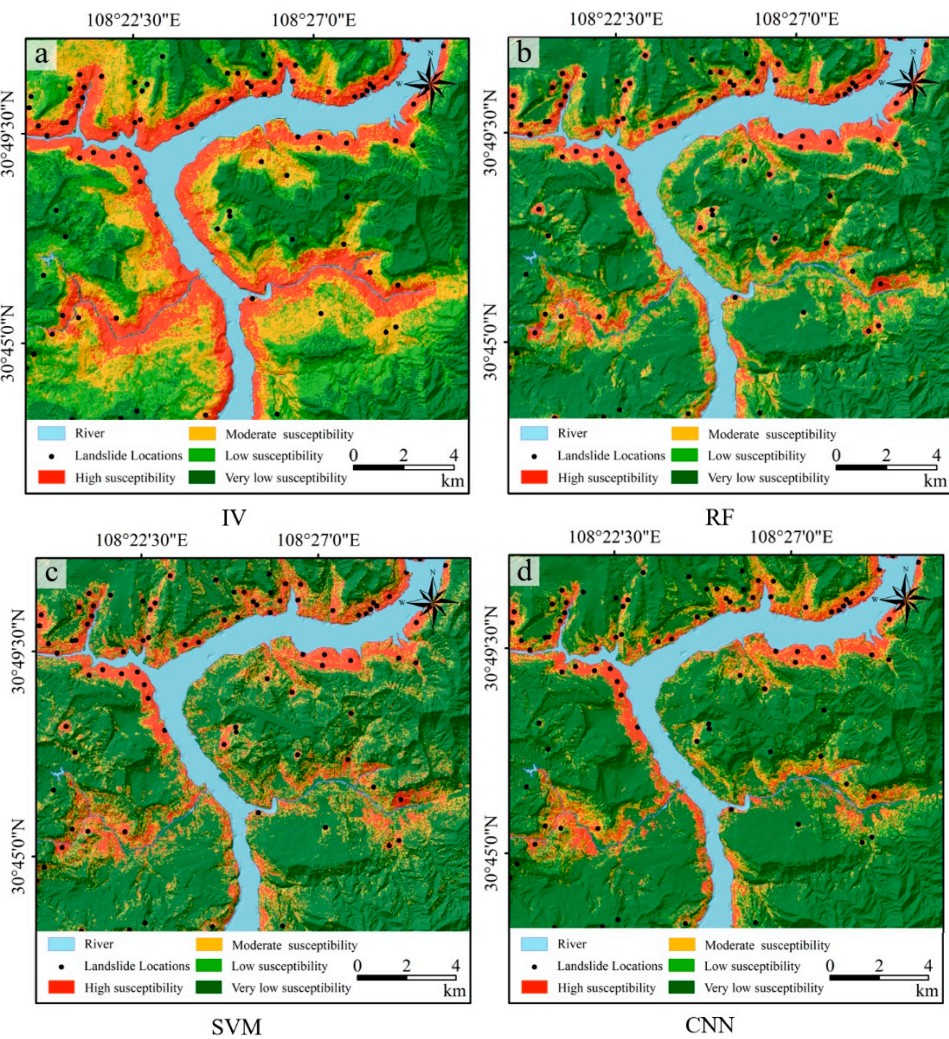

**Figure 5.** Landslide susceptibility maps: (**a**) LSM by IV, (**b**) LSM by RF, (**c**) LSM by SVM, and (**d**) LSM by CNN.

Among these models, the LSM of the IV model demonstrates a strong correlation with the factor of distance to rivers. All areas along the Yangtze River and its tributaries are identified as H susceptibility landslide areas. Overall, the susceptibility tends to decrease as the distance to rivers increases. While the IV model effectively partitions susceptibility areas to some extent, it struggles to distinguish varying landslide susceptibility when regions have similar distances to rivers, primarily due to the high weighting of the distance to rivers factor, which results from the predominant distribution of landslides along watercourses. Therefore, the IV model is considered to have the issue of insufficient differentiation. Table 3 illustrates the landslide grid ratios and information value in each class of IV model.

**Table 3.** Detailed parameters of each class in the IV model.

| Assessment Factors | Value | Landslide Grids | | Total Study Area | | Information Value |
|---|---|---|---|---|---|---|
| | | Count | Percentage (%) | Count | Percentage (%) | |
| Elevation (m) | 149~250 | 5413 | 54.54 | 80,747 | 36.87 | 0.708 |
| | 250~450 | 3710 | 37.38 | 129,436 | 43.06 | −0.142 |
| | 450~650 | 802 | 8.08 | 83,535 | 27.79 | −1.235 |
| | 650~825 | 0 | 0.00 | 6846 | 2.28 | −5.421 |
| Slope (°) | 0~10 | 4077 | 41.08 | 129,857 | 43.20 | −0.505 |
| | 10~20 | 3508 | 35.35 | 92,705 | 30.84 | 0.136 |
| | 20~30 | 1988 | 20.03 | 58,592 | 19.49 | 0.027 |
| | 30~40 | 324 | 3.26 | 16,742 | 5.57 | −0.534 |
| | >40 | 28 | 0.28 | 2668 | 0.89 | −1.146 |
| Slope aspect | Flat | 2 | 0.02 | 20,105 | 6.69 | −5.805 |
| | North | 2040 | 20.55 | 52,729 | 17.54 | 1.588 |
| | Northeast | 1664 | 16.77 | 34,763 | 11.75 | 0.371 |
| | East | 1017 | 10.25 | 29,637 | 9.86 | 0.039 |
| | Southeast | 997 | 10.05 | 28,477 | 9.47 | 0.059 |
| | South | 1304 | 13.14 | 32,859 | 10.93 | 0.184 |
| | Southwest | 670 | 6.75 | 33,016 | 10.98 | −0.487 |
| | West | 995 | 10.03 | 32,163 | 10.70 | −0.065 |
| | Northwest | 1236 | 12.45 | 36,815 | 12.25 | 0.017 |
| Plan curvature | −1< | 4581 | 46.16 | 129,102 | 42.95 | 0.072 |
| | −1~1 | 539 | 5.43 | 35,895 | 11.94 | −0.788 |
| | >1 | 4805 | 48.41 | 135,567 | 45.10 | 0.071 |
| Profile curvature | −1< | 4506 | 45.40 | 129,112 | 42.96 | 0.055 |
| | −1~1 | 355 | 3.58 | 30,525 | 10.16 | −1.044 |
| | >1 | 5064 | 51.02 | 140,927 | 46.89 | 0.085 |
| Lithologic | $J_{3s}$ | 514 | 5.18 | 66,640 | 22.17 | −1.45 |
| | $J_{2s}$ | 9411 | 94.82 | 227,522 | 75.7 | 0.225 |
| | $J_{2xs}$ | 0 | 0.00 | 1174 | 0.39 | −3.658 |
| | $J_{3p}$ | 0 | 0.00 | 5083 | 1.69 | −5.123 |
| | $J_{2x}$ | 0 | 0.00 | 145 | 0.05 | −1.566 |
| Distance to river (m) | 0 | 10 | 0.10 | 30,762 | 10.23 | −4.62 |
| | 0~100 | 4527 | 45.61 | 37,441 | 12.46 | 1.298 |
| | 100~200 | 2870 | 28.92 | 42,636 | 14.19 | 0.712 |
| | 200~300 | 1316 | 13.26 | 71,582 | 23.82 | −0.586 |
| | 300~400 | 915 | 9.22 | 57,638 | 19.18 | −0.732 |
| | >400 | 287 | 2.89 | 60,505 | 20.13 | −1.940 |
| Distance to road (m) | 0~20 | 1036 | 10.44 | 19,987 | 6.65 | 0.451 |
| | 20~50 | 1515 | 15.26 | 27,384 | 9.11 | 0.516 |
| | 50~100 | 1740 | 17.53 | 35,841 | 11.92 | 0.385 |
| | 100~200 | 2146 | 21.62 | 46,378 | 15.43 | 0.337 |
| | >200 | 3488 | 35.14 | 170,974 | 56.88 | −0.482 |
| TWI | 0~5 | 891 | 8.98 | 37,106 | 12.35 | −0.319 |
| | 5~7 | 3458 | 34.84 | 118,051 | 39.28 | −0.112 |
| | 7~10 | 4595 | 46.3 | 96,035 | 31.95 | 0.371 |
| | >10 | 981 | 9.88 | 49,372 | 16.43 | −0.508 |
| NDVI | 0~0.4 | 810 | 8.16 | 46,904 | 15.61 | −0.648 |
| | 0.4~0.6 | 1408 | 14.19 | 32,155 | 10.70 | 0.282 |
| | 0.6~0.75 | 1690 | 17.03 | 35,953 | 11.96 | 0.353 |
| | 0.75~0.9 | 3769 | 37.97 | 100,451 | 33.42 | 0.128 |
| | >0.9 | 2248 | 22.65 | 85,101 | 28.31 | −0.223 |
| Land use type | Agricultural land | 6377 | 64.25 | 171,660 | 57.11 | 0.118 |
| | Forest | 198 | 1.99 | 16,952 | 5.64 | −1.039 |
| | Shrubland | 77 | 0.78 | 4783 | 1.59 | −0.718 |
| | River | 171 | 1.72 | 25,521 | 8.49 | −1.59 |
| | Artificial surface | 3102 | 31.25 | 81,648 | 27.16 | 0.140 |

The susceptibility results of the other three prediction models showed that the areas with H susceptibility were consistent with the landslide distribution area, but there were still some differences in the details. The RF model achieved its best performance with parameters set at max_depth = 20, min_samples_leaf = 1, min_samples_split = 6, and n_estimators = 46. Among the factors, the distance to rivers carried the highest weight of 0.225, while lithology had the lowest weight of 0.028. The LSM generated based on RF-predicted probabilities shows adequate continuity and a reasonable distribution at different susceptibility levels. The SVM model performed best with parameters set at C = 1, kernel = 'rbf', and gamma = 0.8. The susceptibility partitions in the LSM produced by the SVM have a similar distribution to the RF model overall. However, the results of the SVM exhibit fragmentation and lower continuity. The CNN model achieved optimal results with parameters set at learning_rate = 0.01 and batch_size = 64. The LSM demonstrates moderate continuity, and areas closer to watercourses were accurately identified as high susceptibility areas. However, some landslide areas far from watercourses were not correctly labeled as high susceptibility areas.

### 4.3. Model Accuracy Verification

Various statistical model evaluation metrics enable a comprehensive assessment of model accuracy from different perspectives, facilitating a quantitative performance comparison. The ROC curve is a valuable tool for assessing model performance, and the AUC value can provide an intuitive representation of predictive accuracy. According to Figure 6, the AUC values for the IV, SVM, RF, and CNN models are 0.865, 0.953, 0.980, and 0.888, respectively. All models have AUC values above 0.8, indicating their competence in landslide prediction. Notably, the RF and SVM models outperform the others in terms of the AUC value, with the RF model achieving the highest AUC value of 0.980.

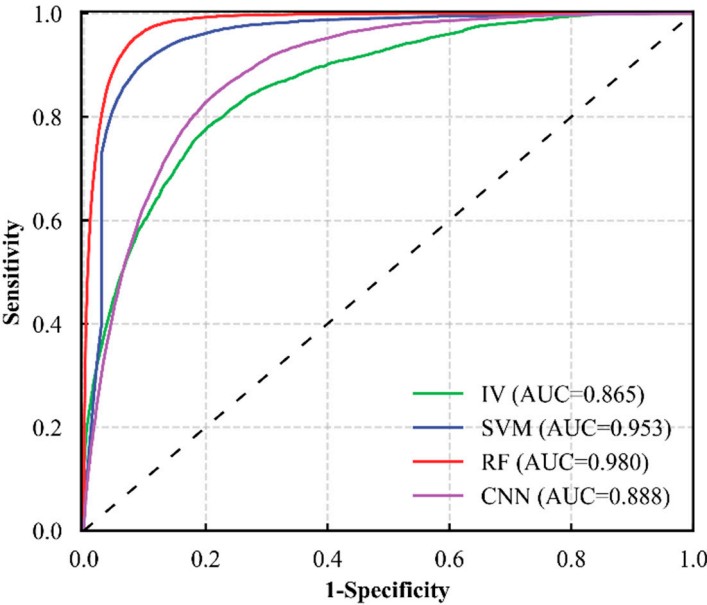

**Figure 6.** ROC curve and AUC values of the four susceptibility models.

The confusion matrix is used to compare the classification results with the actual circumstances and measure the classification performance of models by deriving the parameters of accuracy, recall, precision and F1 score. Models with higher accuracy, F1 score, and a balance in recall and precision are typically regarded as performing better. Figure 7 portrays the accuracy, recall, precision, F1 score, and training speed of the selected models, respectively. For the sake of convenient comparison, the training speed is represented by the natural logarithm of the training time (except for the IV model). As shown in Figure 7, most models perform well in accuracy but show subpar performance in precision. The

RF model exhibited the best performance metrics, with an accuracy of 0.976, a recall of 0.647, a precision of 0.625, an F1 score of 0.636, and the shortest training time compared to the other ML models. These indicators suggest that the model performance is ranked as: RF > SVM > CNN > IV.

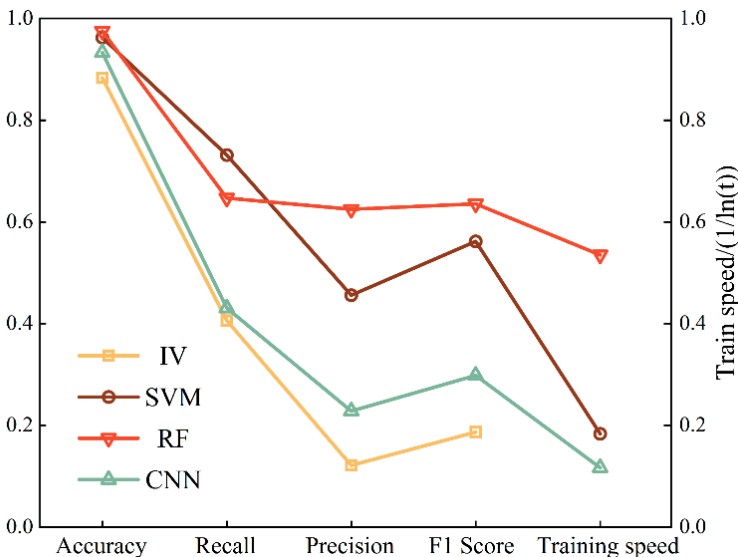

**Figure 7.** Comparison analysis of the applied models based on confusion matrix-derived parameters and training speed.

FR is a reliable technique for quantifying the ability to express landslides within different susceptibility zones. To mitigate the differences in magnitudes and facilitate a more convenient comparison, natural logarithm processing was applied to FR, and the results are presented in Figure 8. Models with a higher FR in the H susceptibility zone and a lower FR in the M, L, and VL susceptibility zones are regarded as possessing enhanced classification capabilities. As shown in Figure 8, the prediction results of the SVM and RF exhibit higher landslide raster ratios in the H susceptibility zone and lower values in other susceptibility zones. In contrast, the IV and CNN models show lower landslide raster ratios in the H susceptibility zone but higher values in other susceptibility zones. The results indicate the superior ability of SVM and RF to distinguish landslide-prone areas.

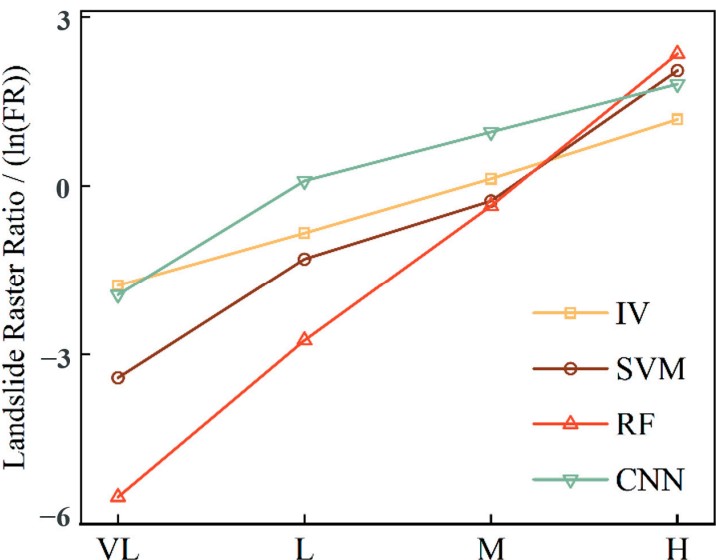

**Figure 8.** Landslide raster ratio analysis and comparison based on natural logarithmically scaled FR.

Based on the comparative analysis conducted, it is evident that the SVM model and RF model exhibit the best performance in analyzing the relationship between landslide disasters and landslide-related factors, providing the most accurate landslide susceptibility predictions in the study area. Consequently, we have selected the SVM and RF models as the foundation for coupling with InSAR deformation data and conducting further analysis.

### 4.4. Result of PS-InSAR

Based on the SAR dataset, persistent scatterer interferometry (PSI) was conducted in ENVI 5.6. The scenes were paired to create 23 connection graphs, and the baseline is illustrated in Figure 9. Subsequently, interferometric processing, two inversions, and geocoding were performed, resulting in a deformation dataset containing 260,039 PS points. The parameters used in the PS process are detailed in Table 4.

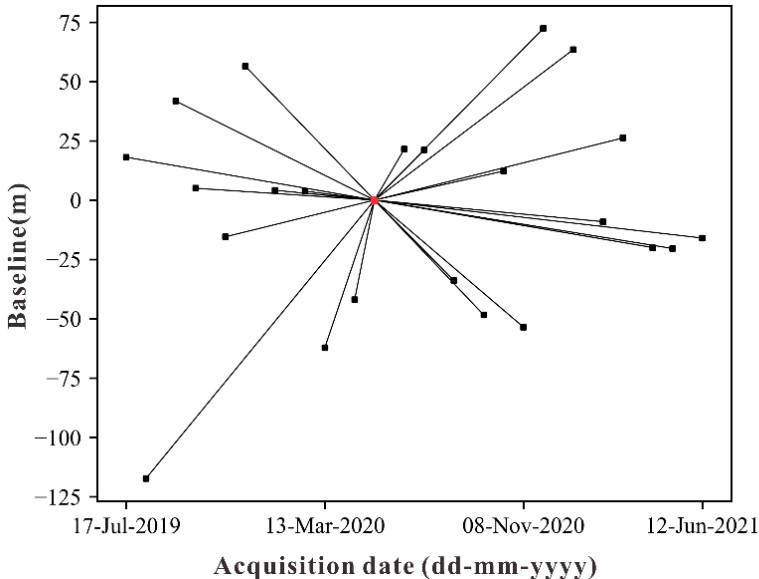

**Figure 9.** Baseline distribution of Sentiel-1A acquisitions.

**Table 4.** Main set parameters for PS processing.

| Parameter | Value |
| --- | --- |
| Orbit configuration | Ascending |
| Size of scenes | $40 \times 85$ km |
| Number of scenes | 24 |
| Look azimuth-angle | 80.45° |
| Max. temporal baseline | 396 days |
| Max. normal baseline | 119.23 m |
| Coherence thresholds | 0.35 |
| Subarea for single reference point | 25 km$^2$ |
| Overlap for subarea | 30% |

After removing noisy PS points, we converted the PS points into a raster image in ArcGIS using their deformation rates as a field, producing the deformation rates in PS points for the study area (as shown in Figure 10a). Then, the "Kriging" tool was employed to interpolate the PS points, generating the deformation rates for the entire study area, as depicted in Figure 10b.

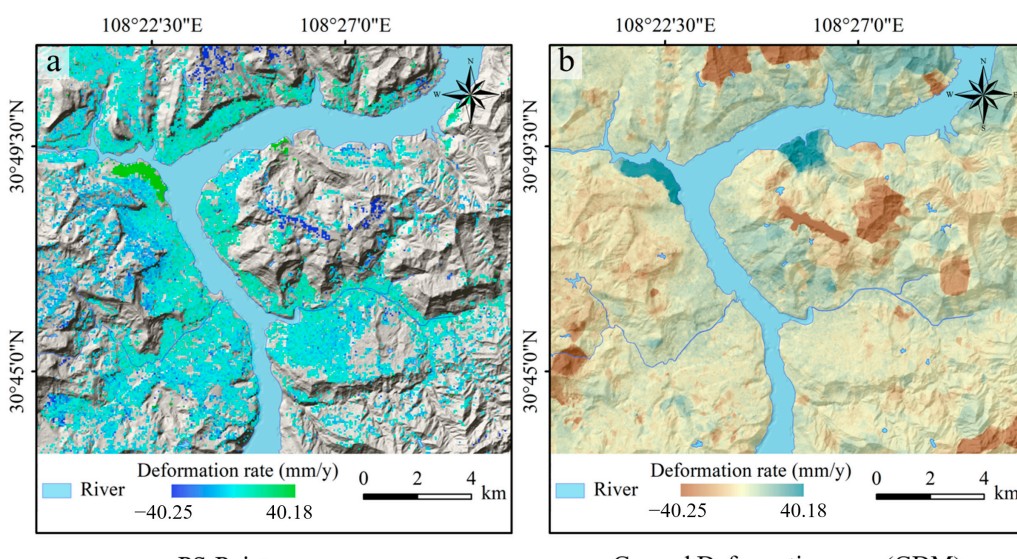

**Figure 10.** (**a**) Mean vertical deformation rate PS-points map; (**b**) Ground deformation map of the study area after Kriging.

As shown in Figure 10, the areas adjacent to rivers in the study area exhibit an uplift trend, while the low hills and ridges regions further from rivers display subsidence trends. The maximum annual average uplift rate within the study area, measured at 40.18 mm/year, was observed in the urban area on the south bank of the Zhuxi River. Conversely, the highest subsidence rate, reaching 40.25 mm/year, was identified around Wanzhou Wuqiao Airport and the hilly region north of the Yangtze River.

*4.5. Landslide Dynamic Susceptibility Mapping*

The SVM and RF models were coupled with InSAR in two different methods, resulting in four landslide dynamic susceptibility assessment models: InSAR weighted overlay SVM (IWSVM), InSAR jointly trained SVM (IJSVM), InSAR weighted overlay RF (IWRF), and InSAR jointly trained RF (IJRF). The landslide probability predictions for each model were segmented through the natural breakpoint method in ArcGIS and visualized as LDSM (as shown in Figure 11). Overall, the LDSMs of the coupled models show a similar susceptibility distribution to the uncoupled models. With the introduction of the deformation characteristic factor, the jointly trained models demonstrate a decrease in M and L susceptibility areas, while VL susceptibility areas increase; in the weighted overlay models, the M and L susceptibility areas increase and VL susceptibility areas decrease.

The LDSM of the IJSVM model has partially alleviated the issue of fragmentation in susceptibility partitions. However, there is still some intermingling in certain H and M susceptibility areas, resulting in insufficient continuity. The IWSVM model predicts more areas as H susceptibility areas, with more dispersion-prone areas. The IJRF model achieved a more reasonable distribution of susceptibility partitions, accurately identifying landslide areas as H susceptibility areas. The susceptibility of non-landslide areas decreased after considering the deformation information feature, leading to a reduction in H, M, and L susceptibility areas and an increase in VL susceptibility areas. The prediction results of the IWRF model exhibit more M and L susceptibility areas and a concentration of H susceptibility areas around the watercourses.

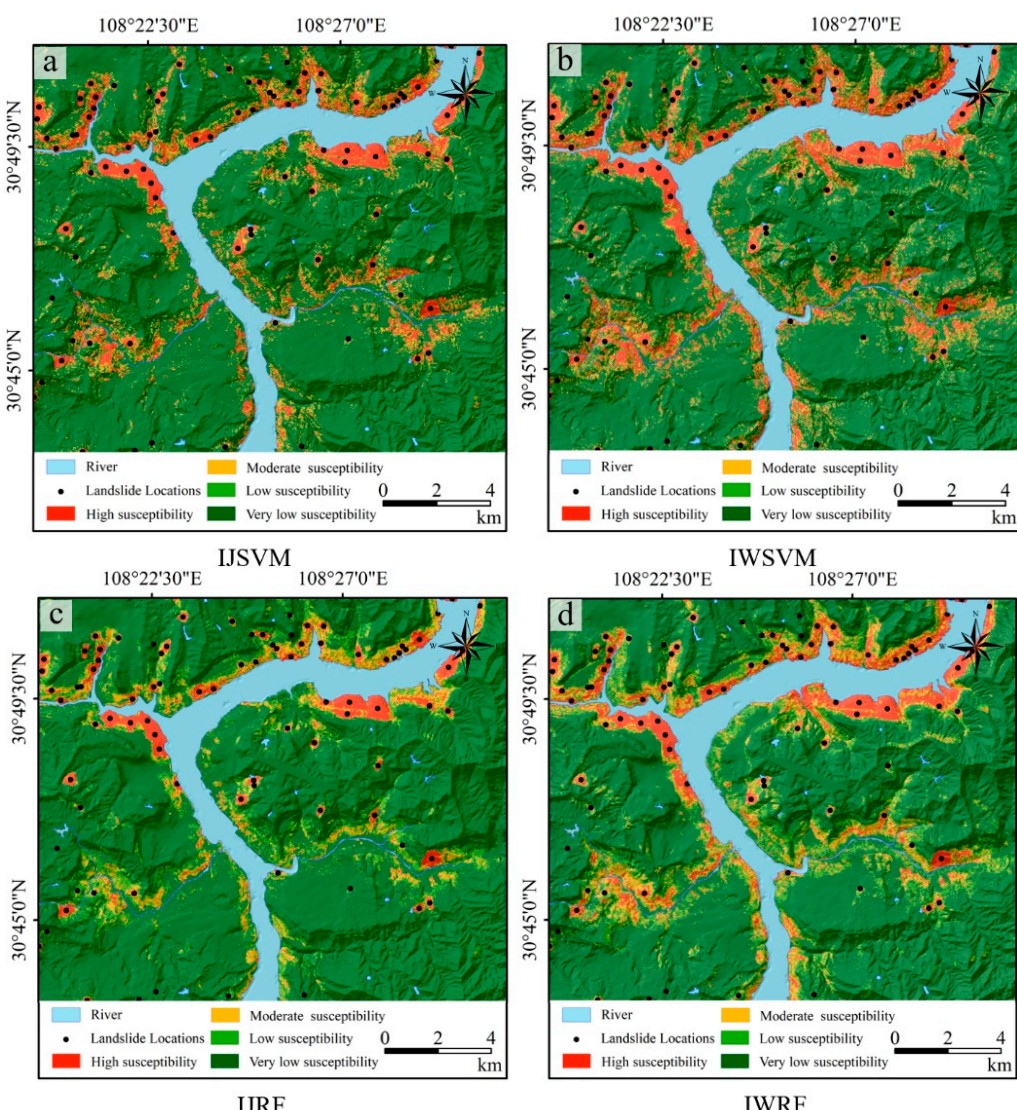

**Figure 11.** Landslide dynamic susceptibility maps generated by coupled models: (**a**) LDSM by IJSVM, (**b**) LDSM by IWSVM, (**c**) LDSM by IJRF, and (**d**) LDSM by IWRF.

## 5. Discussion

Landslide susceptibility mapping is a crucial method in the risk assessment and control of landslide disasters. However, due to the lack of dynamic landslide characteristics, which results in the absence of landslide movement state features, traditional models fail to provide reliable susceptibility assessment results. In our previous research, four models were applied for landslide susceptibility mapping in the study area, and a comparative analysis of these models was conducted using various evaluation parameters. Then, we used PS-InSAR technology to obtain the GDM of the study area and coupled the GDM with the SVM and RF models using two methods, resulting in four dynamic models and the generation of LDSMs for the study area. The difference between the LDSM and LSM indicates the effect of InSAR deformation data, demonstrating the feasibility of susceptibility assessment models that consider dynamic features. However, different coupling methods expressed the InSAR deformation data in separate ways. Therefore, we employed the same evaluation methods as before, validating the performance of these dynamic susceptibility models. Through further comparative analysis, we aimed to evaluate the performance of different dynamic models and assess the effect of InSAR deformation data in these models.

### 5.1. Performance Comparison of Landslide Dynamic Susceptibility Models

The original models and the four dynamic models before and after considering dynamic features were evaluated based on ROC curves and AUC values. The comparative results are illustrated in Figure 12. The AUC values for IJRF, IWRF, IJSVM, and IWSVM were 0.995, 0.973, 0.972, and 0.947. IJRF and IJSVM exhibited improvements of 1.53% and 1.99% compared with RF and SVM. Conversely, the AUC values of IWRF and IWSVM showed reductions of 0.71% and 0.63%. Among these models, the IJRF model achieved the best performance, with an AUC value of 0.995. The results of the ROC indicated that the coupling method of joint training enhances model accuracy, whereas the weighted overlay method diminishes model accuracy.

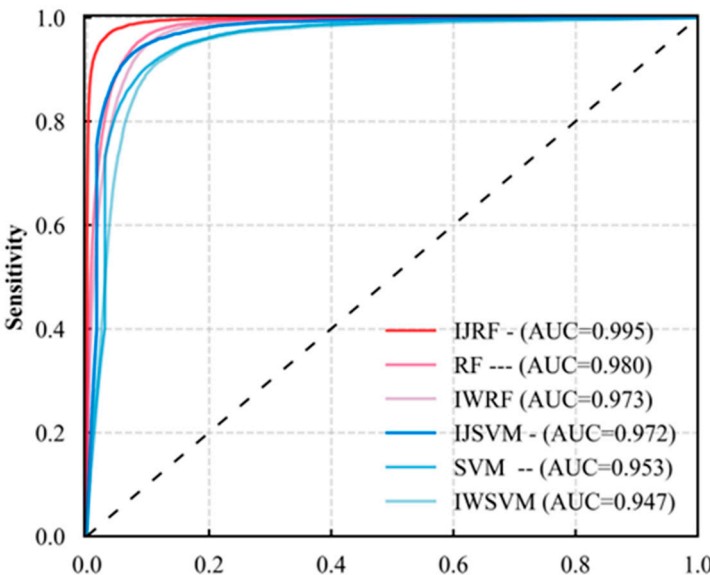

**Figure 12.** ROC curves and AUC values of RF, SVM, and landslide dynamic susceptibility models.

Similarly, the computed confusion matrix derived further parameters of dynamic susceptibility assessment models, as shown in Figure 13. It is evident from the figure that all models exhibit an accuracy exceeding 0.95, indicating their high precision. However, except for IJRF, the precision values of the models are significantly lower than their recall values, indicating that these models fail to strike a balance between recall and precision. This suggests these models could predict landslides but struggle in ensuring the accuracy of samples identified as landslides. In contrast, the IJRF model displays an accuracy, recall, precision, and F1 score of 0.991, 0.847, 0.885, and 0.865, respectively, and has the highest performance of all examined models in these terms.

FR values represent the landslide proportions in the VL, L, M, and H susceptibility levels for each model. These FR values are naturally logarithmically scaled and are presented in Figure 14. From the figure, it can be observed that SVM and its coupled models have higher landslide proportions in the VL and L susceptibility areas but lower proportions in the H susceptibility areas. That indicates its poor discrimination ability in distinguishing landslide susceptibility areas. On the other hand, the RF and its coupled models have both lower landslide proportions in the VL and L susceptibility areas and higher proportions in M and H partitions. Notably, the IJRF model, while having a lower proportion of landslide areas in the VL susceptibility area, has the highest proportion of landslide areas in the H susceptibility area, indicating the superiority of the IJRF model in discriminating landslide susceptibility areas from non-landslide susceptibility areas.

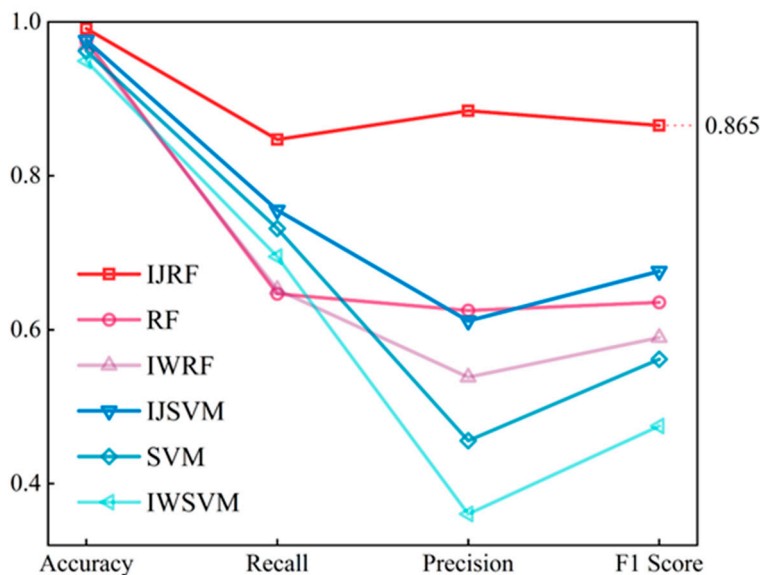

**Figure 13.** Comparison analysis of RF, SVM, and dynamic models based on confusion matrix-derived parameters.

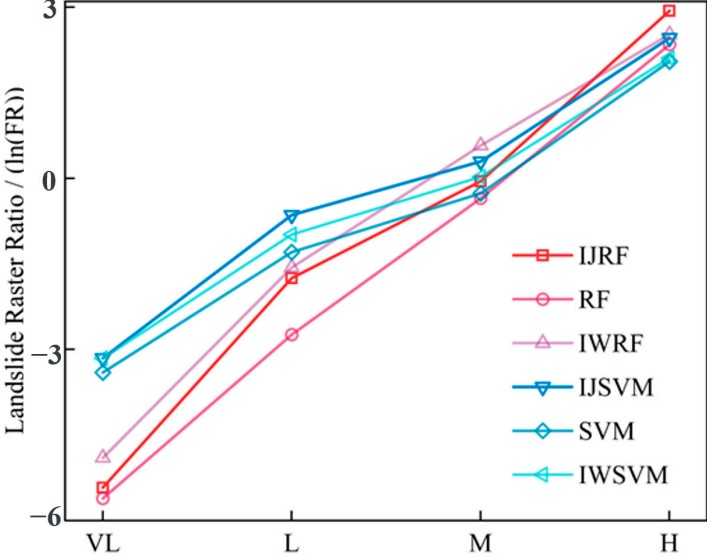

**Figure 14.** Landslide raster ratio analysis and comparison of dynamic model based on natural logarithmically scaled FR.

### 5.2. Effect Analysis of InSAR Deformation Data

The importance of landslide-related factors in the RF and IJRF models is depicted in Figure 15. Factor importance indicates the extent to which each factor influences the model's classification performance, signifying how each factor helps the model explain the relationships of landslides and related factors. According to Figure 15, these two models share a similar distribution of factor importance, with the distance to rivers being the most crucial factor in both cases, while lithology, distance to roads, land use type have lower factor importance. After the inclusion of InSAR deformation data as a factor, the importance of other factors in the IJRF model slightly decreased, with InSAR deformation data having an importance score of 0.154, ranking third among the twelve factors. In the IJRF model, InSAR deformation data as a factor provides discriminative information between landslide and non-landslide samples, playing a significant role in the RF model's classification decisions, thereby enhancing the model's performance.

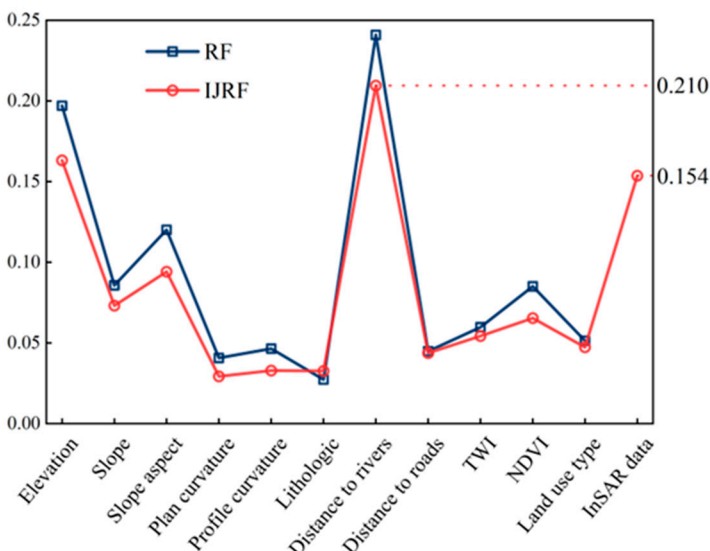

**Figure 15.** The importance of factors in the RF and IJRF.

Furthermore, we discuss the impact of introducing InSAR deformation data by analyzing the changes in LSM within typical landslide areas. The Sifangbei landslide and Cizhuyuan landslide areas were chosen as representative cases; their LSMs generated by RF and GDM and LDSMs obtained through IJRF are depicted in Figure 16.

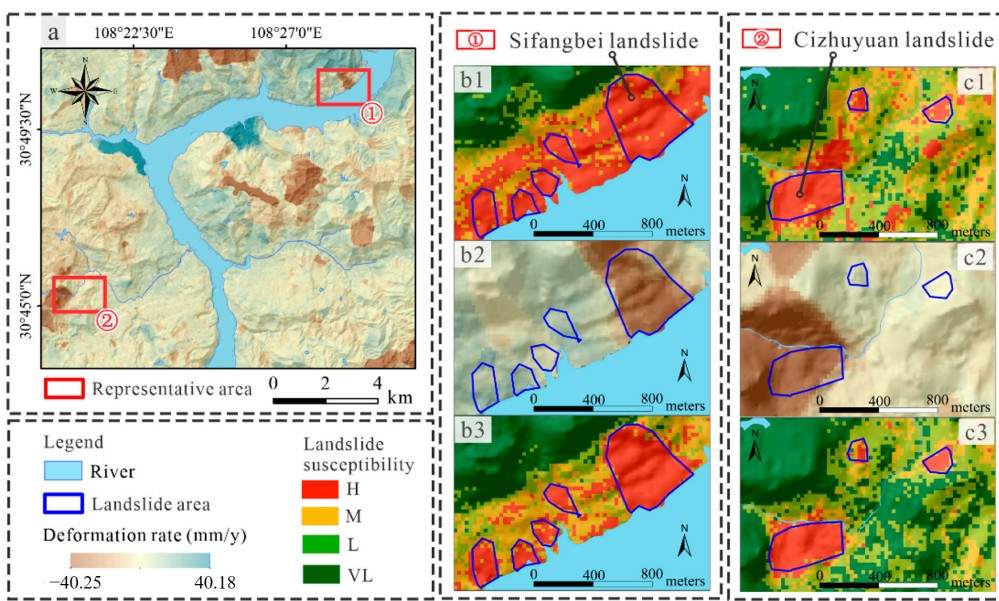

**Figure 16.** Detailed comparative analysis of typical landslide areas (**a**) based on the LSM, GDM, and LDSM of the Sifangbei Landslide area (**b1**–**b3**) and the Cizhuyuan Landslide area (**c1**–**c3**).

For the Sifangbei landslide area, situated on the northeast bank of the Yangtze River (a), the RF model identified the riverbank portion as a high susceptibility area for landslide disasters, but the LSM (b1) failed to accurately delineate the landslide boundaries. The GDM (b2) revealed substantial subsidence deformation in the Sifangbei landslide area compared to its surroundings. The inclusion of InSAR deformation data in the IJRF model led to improved differentiation between landslide and non-landslide areas along the riverbank area, as evidenced by the LDSM (b3). It effectively outlined the regions of the Sifangbei landslide while reducing the susceptibility of other surrounding non-landslide areas, enhancing the model's performance.

The Cizhuyuan landslide area is located at the source of the Longbao River on the west bank of the Yangtze River (a). In this area, the LSM generated by the RF model (c1) predicted a significant number of false-positive areas within the H and M susceptibility zones. The introduction of the InSAR deformation factor (c2) reduced the H and M susceptibility areas for non-landslide areas in the LSM, resulting in a more accurate judgment of non-landslide regions in the LDSM (c3).

Overall, these comparisons demonstrate the enhanced performance of the IJRF model with the consideration of deformation features, which aids in more accurately identifying landslide areas and reducing susceptibility ratings for non-landslide regions around the landslide susceptible area.

## 6. Conclusions

Remote sensing images and the time-series InSAR technique provide critical data for the early identification and stability assessment of landslides [31,68,69], but are often under-utilized in existing landslide susceptibility evaluation studies. Focusing on the Wanzhou urban area within the Three Gorges Reservoir area (TGRA), this paper presented landslide susceptibility mapping (LSM) and time series deformation analysis based on geological and Sentinel-1 data and explored the landslide dynamic susceptibility mapping (LDSM) method coupled with machine learning and the PS-InSAR model in different approaches.

LSM shows that high landslide susceptibility areas in the Wanzhou urban area are primarily concentrated along the riverside. The SVM and RF show superior performance in various evaluation indicators than other models (IV, CNN). The subsidence of Wanzhou urban area obtained using InSAR techniques ranges from $-40.25$ to $40.18$ mm/year, with some deformed regions correlated with potential landslides. After incorporating InSAR deformation data as a training factor, the results of the IJRF model exhibited fewer false positives and a more precise delineation of landslide areas. Additionally, the importance of factors indicated the critical role of InSAR deformation data in the IJRF model.

Summarily, the integration of time series InSAR analysis with the RF machine learning model effectively enhanced the LSM model's precision, contributing to disaster prevention and mitigation in the TGRA. However, the C-band (wavelength 5.6 cm) data used in this study was prone to decorrelation with the presence of vegetation, leading to inaccurate estimations of ground deformation [32]. Therefore, longer-band, multi-spatial view SAR data and multi-temporal InSAR (MT-InSAR) methods will be considered to obtain comprehensive ground deformation data and assist in landslide susceptibility mapping.

**Author Contributions:** Writing—original draft, F.M.; Data curation, Q.R.; Writing—review and editing, Y.W. and Z.Q. (Zhao Qian); Methodology, Z.K.; Investigation, Z.Q. (Zhangkui Qin). All authors have read and agreed to the published version of the manuscript.

**Funding:** This research was supported by the National Key R&D Program of China (2023YFC3007201), National Natural Science Foundation of China (42377161, 41977244), Natural Science Foundation of Hubei Province (2023AFB580), and Guizhou Provincial Science and Technology Project (QKHZC[2023]YB127). The authors thank the colleagues in our laboratory for their constructive comments and assistance.

**Institutional Review Board Statement:** Not applicable.

**Informed Consent Statement:** Not applicable.

**Data Availability Statement:** This study are available on request from the corresponding author.

**Conflicts of Interest:** The authors declare no conflict of interest.

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
