# Peer review of "Landslide Dynamic Susceptibility Mapping Base on Machine Learning and the PS-InSAR Coupling Model"

_remotesensing, doi:10.3390/rs15225427_

Round 1

Reviewer 1 Report

Comments and Suggestions for Authors

In this research, based on Sentinel-1 data, relevant data and existing research results, IV, RF, SVM, and CNN models were selected to analyze landslide susceptibility in the urban area of Wanzhou. Models with superior performance will be coupled with PS-InSAR deformation data using two methods: joint training and weighted overlay. The accuracy of different models was assessed and compared, with the aim of determining the optimal coupling model and the role of InSAR in the model. I think the topic discussed in this paper is very important, which is of great significance for research of landslide susceptibility mapping. This reviewer sees that a minor revision will be needed before being accepted for possible publication. Here are the main comments for the revision.

(1) A key problem in this paper is the lack of introduction to the applicability of the proposed method. Is this model applicable to all region?

(2) The title of the paper mentions “Landslide dynamic susceptibility mapping”. What is “dynamic”? What is the difference from traditional susceptibility mapping?

(3) What is the relationship between susceptibility map and ground deformation map in Figure 1?

(4) Did the author consider the correlation between various indicators in the selection of susceptibility evaluation factors? Are they independent of each other?

(5) Figure 10 shows that the AUC values of each model are very high. Does this mean that there is overfitting in the model?

(6) In Figure 8, did the ground deformation map obtained by Kriging interpolation consider the influence of terrain and topography? In addition, which direction is the surface deformation obtained from remote sensing interpretation?

(7) The conclusion is not concise and innovative. I believe that the Authors should try to interpret and explain more clearly their results. Some key quantitative conclusions should be supplemented.

(8) What are the shortcomings of this study?

Author Response

We do owe so much to the editor for giving us opportunity to improve the manuscript entitled “Landslide dynamic susceptibility mapping base on machine learning and PS-InSAR coupling model”. Likewise, we also want to express our sincere thanks to 4 anonymous reviewers and the editor, who contributed their time, effort, and patience in reading the manuscript conscientiously and providing some insightful comments. We have studied each comment and have made correction point by point carefully, which we hope meet with approval. Revised portion are marked in red in the paper. The responses to each comment are listed in Response to Reviewers.

Reviewer 2 Report

Comments and Suggestions for Authors

This study conduct the Landslide dynamic susceptibility mapping base on machine  learning and PS-InSAR coupling model. The most popular method including IV, RF, SVM, and CNN models are used for LSM. And then The PS-InSAR result used as the influencing factor for mapping. Finally, the two models are compared for studying the effect of InSAR result on landslide susceptibility mapping.

1. The flowchart is not proper in the Introduction section. It should be in the Section 3.

2. The English writing should be improved. Some errors exists in the text.

3. The study area should be section 2.

4. Why did you used formula 12.

5. The scale of the study area is small. The significance of the study is difficult to convince.

6. From my perspective, The added the InSAR result did not increase the accuracy of the modelling. The little increase of AUC is not conviced.

7. The information about data and calculation process of PS-InSAR should be added.

8. The training set and validation set should be separated.

9. The conclusion is too long.

10. The significance and the purpose of the study should be clarified.

11. The novelty of the study is not clear.

Comments on the Quality of English Language

This study conduct the Landslide dynamic susceptibility mapping base on machine  learning and PS-InSAR coupling model. The most popular method including IV, RF, SVM, and CNN models are used for LSM. And then The PS-InSAR result used as the influencing factor for mapping. Finally, the two models are compared for studying the effect of InSAR result on landslide susceptibility mapping.

1. The flowchart is not proper in the Introduction section. It should be in the Section 3.

2. The English writing should be improved. Some errors exists in the text.

3. The study area should be section 2.

4. Why did you used formula 12.

5. The scale of the study area is small. The significance of the study is difficult to convince.

6. From my perspective, The added the InSAR result did not increase the accuracy of the modelling. The little increase of AUC is not conviced.

7. The information about data and calculation process of PS-InSAR should be added.

8. The training set and validation set should be separated.

9. The conclusion is too long.

10. The significance and the purpose of the study should be clarified.

11. The novelty of the study is not clear.

Author Response

(The authors gave the same response as above.)

Reviewer 3 Report

Comments and Suggestions for Authors

The paper titled "Landslide Dynamic Susceptibility Mapping Base on Machine Learning and PS-InSAR Coupling Model" addresses a critical issue in landslide susceptibility assessment using a combination of machine learning models (IV, RF, SVM, CNN) and PS-InSAR deformation data. The study focuses on the Three Gorges Reservoir area, emphasizing the dynamic nature of landslides and the importance of accurate susceptibility mapping for risk management. The language used in the paper is generally clear and technical, catering to an audience well-versed in remote sensing and machine learning. Here's a detailed review:

Strengths:

Clear Problem Statement: The paper effectively outlines the problem of landslide susceptibility mapping in the Three Gorges Reservoir area, providing context for the study's significance.

Comprehensive Methodology: The paper describes a thorough methodology involving the use of various machine learning algorithms and PS-InSAR data, demonstrating a systematic approach to tackle the issue at hand.

Detailed Results Presentation: The results are presented in a detailed manner, including figures and comparative analyses. The use of ROC curves, AUC values, and confusion matrices enhances the paper's technical rigor.

In-depth Discussion: The discussion section delves into the nuances of different models and coupling methods, providing a comprehensive understanding of the results. The analysis of factors and their importance in model performance adds depth to the discussion.

Comparative Examples: The inclusion of comparative examples, such as the Sifangbei and Cizhuyuan landslide areas, effectively illustrates the impact of InSAR deformation data on model accuracy.

Areas for Improvement:

Literature Review: It can be made more comprehensive. This is a global problem and you can cite different studies. Such as 

Rules Reservoir, Spain (https://link.springer.com/article/10.1007/s10346-021-01728-z)

Baglihar Reservoir , India (https://www.sciencedirect.com/science/article/pii/S0926985122002257)

Potrerillos Dam Reservoir, Argentina (https://link.springer.com/article/10.1007/s10346-015-0583-4)

Chin Coulee Reservoir, Canada (https://www.mdpi.com/2072-4292/13/3/366)

 Mazar Dam, Ecuador (https://link.springer.com/article/10.1007/s10346-022-01913-8)

also, these studies substantiate the author's idea of using InSAR data for Landslide mapping.

Clarity in Language: While the technical language is suitable for the target audience, there are instances where sentences could be more concise and clearer. For example, in the conclusion, the sentence "Traditional landslide susceptibility assessment models, however, doe to the neglect of surface dynamic characteristics, leading to false-negative errors in the result." seems to contain a typographical error ("doe" should be "due"). Proofreading for such errors can enhance the paper's readability.

Also, there is an error in the title itself. 

Experiment Information: The authors should provide more data on InsAR processing (no and dates of images in appendix), threshold parameters and reference point etc. The paper should enable readers to repeat the experiment.

Contextualization in Certain Sections: In some parts of the paper, especially the conclusion, certain concepts are assumed to be understood without adequate explanation. Providing a bit more context or background information can help readers who might not be intimately familiar with every aspect of the topic.

Visualization Enhancement: While the paper describes figures (e.g., Fig. 14) depicting the results, a bit more context within the figures or captions could aid readers in understanding the significance of the visualized data. Explaining what specific features indicated within the figures would be helpful.

Correction in Figures: In Figure 14, meters should be written all in small letters.

Future Work and Limitations: The conclusion lacks a discussion on the limitations of the study and suggestions for future research. Including these aspects would provide a more holistic view of the research and its potential directions.

In summary, the paper presents a robust study with valuable findings in the field of landslide susceptibility mapping. With some attention to language clarity, contextualization, visualization enhancement, and addressing limitations, the paper could further enhance its impact and accessibility to a wider readership.

Comments on the Quality of English Language

English editing is needed.

Author Response

(The authors gave the same response as above.)

Reviewer 4 Report

Comments and Suggestions for Authors

This article discusses research about landslide dynamic susceptibility mapping base on machine learning and PS-InSAR coupling model. I find this paper is interesting, which is suitable for publication in Remote Sensing. There are some questions that need to be addressed before being accepted.

1. Why did the author choose the SVM, CNN, RF models in this paper?

2. Is the vertical displacement or horizontal displacement obtained from remote sensing interpretation? The deformation of landslides should be mainly caused by horizontal displacement. Can the results of remote sensing interpretation be directly applied to the evaluation of landslide susceptibility?

3. Is there any correlation between each factors in this paper? Are they independent of each other?

4. The conclusion section is not concise enough.

5. Figure 8 shows that the ground deformation in the study area is quite obvious. Can the author provide additional explanations on the credibility of remote sensing interpretation results.

6. Legends in Figure 14 are missing.

Above all, this paper requires minor revisions before publication.

Author Response

(The authors gave the same response as above.)

Round 2

Reviewer 3 Report

Comments and Suggestions for Authors

The authors have made necessary changes.

Still, they need to one minor change. The way of citing the references is erroneous.

From line 80 to 85:

"Utilizing the Persistent Scatterer Interferometry (PSI) technique, Vishal  Mishra [32] retrieved the displacements of Baglihar Dam Reservoir slope, Sajid Hussain [33] updated the landslide inventory for susceptibility mapping; Zhifu Zhu[34] constructed the landslide dynamic susceptibility assessment model by means of empirical matrix; Chao Zhou[35] , Chen Cao[36] used the weighted overlay method to construct the landslide dynamic susceptibility assessment model which obtain"

Don't write the name of the first author for citing unless it is a singly authored paper. Here all paper have multiple authors hence either use "A & B [34]" or ''C et al. [35]'

Also, reference no. 35 as per the reference list is 'Liu et al' and in the text corresponding to 35 'Chao Zhou' is mentioned. Correct it

Comments on the Quality of English Language

Way of referencing is in correct.

Author Response

We do owe so much to the editor for giving us opportunity to improve the manuscript entitled “Landslide dynamic susceptibility mapping base on machine learning and PS-InSAR coupling model”. Likewise, we also want to express our sincere thanks to the reviewers and the editor, who contributed their time, effort, and patience in reading the manuscript conscientiously and providing some insightful comments. We have studied each comment and have made correction point by point carefully, which we hope meet with approval. Revised portion are marked in red in the paper. The responses to each comment are listed as follows:

Responds to the reviewer’s comments:

Reviewer #3:

  1. Response to comment: The authors have made necessary changes. Still, they need to one minor change. The way of citing the references is erroneous.

From line 80 to 85:

"Utilizing the Persistent Scatterer Interferometry (PSI) technique, Vishal  Mishra [32] retrieved the displacements of Baglihar Dam Reservoir slope, Sajid Hussain [33] updated the landslide inventory for susceptibility mapping; Zhifu Zhu[34] constructed the landslide dynamic susceptibility assessment model by means of empirical matrix; Chao Zhou[35] , Chen Cao[36] used the weighted overlay method to construct the landslide dynamic susceptibility assessment model which obtain"

Don't write the name of the first author for citing unless it is a singly authored paper. Here all paper have multiple authors hence either use "A & B [34]" or ''C et al. [35]'

Also, reference no. 35 as per the reference list is 'Liu et al' and in the text corresponding to 35 'Chao Zhou' is mentioned. Correct it

Response: Thank you for your requests. The way of citing the references has been revised in the manuscript.

Utilizing the Persistent Scatterer Interferometry (PSI) technique, Mishra & Jain [32] retrieved the displacements of Baglihar Dam Reservoir slope. Hussain et al. [33] updated the landslide inventory for susceptibility mapping. Zhu et al. [34] constructed the landslide dynamic susceptibility assessment model by means of empirical matrix. Liu et al [35] , Cao et al. [36] used the weighted overlay method to construct the landslide dynamic susceptibility assessment model which obtain LDSM (Landslide Dynamic Susceptibility Mapping) by weighting the InSAR deformation data and LSM.
